# Analysis of Copy Number Variation in the Whole Genome of Normal-Haired and Long-Haired Tianzhu White Yaks

**DOI:** 10.3390/genes13122405

**Published:** 2022-12-18

**Authors:** Guangyao Meng, Qi Bao, Xiaoming Ma, Min Chu, Chun Huang, Xian Guo, Chunnian Liang, Ping Yan

**Affiliations:** 1Key Laboratory of Yak Breeding Engineering Gansu Province, Lanzhou Institute of Husbandry and Pharmaceutical Science, Chinese Academy of Agricultural Sciences, Lanzhou 730050, China; 2Key Laboratory of Animal Genetics and Breeding on Tibetan Plateau, Ministry of Agriculture and Rural Affairs, Lanzhou 730050, China

**Keywords:** Tianzhu white yaks, copy number variation, long-haired trait, resequencing

## Abstract

Long-haired individuals in the Tianzhu white yak population are a unique genetic resource, and have important landscape value. Copy number variation (CNV) is an important source of phenotypic variation in mammals. In this study, we used resequencing technology to detect the whole genome of 10 long-haired Tianzhu white yaks (LTWY) and 10 normal-haired Tianzhu white yaks (NTWY), and analyzed the differences of CNV in the genome of LTWYs and NTWYs. A total of 110268 CNVs were identified, 2006 CNVRs were defined, and the distribution map of these CNVRs on chromosomes was constructed. The comparison of LTWYs and NTWYs identified 80 differential CNVR-harbored genes, which were enriched in lipid metabolism, cell migration and other functions. Notably, some differential genes were identified as associated with hair growth and hair-follicle development (e.g., *ASTN2*, *ATM*, *COL22A1*, *GK5*, *SLIT3*, *PM20D1*, and *SGCZ*). In general, we present the first genome-wide analysis of CNV in LTWYs and NTWYs. Our results can provide new insights into the phenotypic variation of different hair lengths in Tianzhu white yaks.

## 1. Introduction

Yak (Bos grunniens) is the main livestock living on the Qinghai–Tibet Plateau and its surrounding areas [1]. It can provide meat, milk, hair and fuel for people in pastoral areas, and is closely related to the life of people in the plateau area [2]. White individuals are rare in yak populations, and coat color is not stably inherited. The Tianzhu white yak is a unique local breed in the Tianzhu Tibetan Autonomous County, Gansu Province, with a large population size. Its coat is white, easy to dye, and can be used to make wool products, which has a very high economic value. In recent years, researchers have found some long-haired individuals in the Tianzhu white yak population, which are characterized by longer hair-length and density on the forehead and body sides than the normal-haired Tianzhu white yak (Figure 1). Based on data from the breeding base, the LTWY has a forehead hair-length of more than 13 cm and the NTWY has a forehead hair-length of less than 13 cm [3]. At present, the number of the Tianzhu white yak population is 69,000, and the number of the long-haired-type core population is more than 600. Because of its unique value in the landscape and its genetic hair resources, studying and breeding long-haired individuals of the white yak is of great significance.

Copy number variation (CNV) is a kind of microscopic and submicroscopic structural variation on the genome, which is usually manifested as the insertion, deletion or repetition of a genome sequence [4,5]. It ranges from 50 bp to larger DNA fragments, and it is an important factor leading to the genetic diversity of organisms [6]. SNP and indel are also important factors leading to biological genetic diversity [7,8]. However, differently from SNP and indel, CNV affects gene expression, individual phenotypic differences and the adaptation of phenotypic changes by changing gene dosage and the gene regulatory region [9]. In the study of important economic traits of livestock and poultry, association studies of SNP have been widely used [10,11,12,13]. However, with the development of molecular biology and the progress of sequencing technology, more and more CNVs have been able to be accurately detected, and the research on CNV has become a research hotspot [14,15,16]. In recent years, researchers have transitioned their research on CNV from genetic diseases related to humans, to diseases and traits of livestock and poultry, and more and more research results show that CNV is closely related to traits of livestock and poultry [17,18].

Previous studies have shown that CNV is associated with important economic traits in livestock. For example, DNA dosage and EST expression of Netrin-1 (*Ntn1*) gene overlapped with CNV region may affect pork quality [19]. In the study of local cattle breeds in China, it was found that CNV of the *MLLT10* gene had positive effects on growth traits [20]. The CNV region can inhibit the expression of the *GBP6* gene, and affect the body-weight traits of cattle [21]. In addition, some studies have shown that CNV of the *ASIP* allele is associated with different coat colors in different breeds of goats and Tibetan sheep [22,23]. Therefore, we can explore the causes of phenotypic differences in Tianzhu white yak hair by studying copy number variation.

In this study, we obtained the whole genome sequence of the Tianzhu white yak, based on whole-genome resequencing technology, and explored the CNV in the genomes of long-haired and normal-haired types of Tianzhu white yak, and explored candidate genes related to the hair length of the Tianzhu white yak.

## 2. Materials and Methods

### 2.1. Animal Welfare

The Lanzhou Institute of Husbandry and Pharmaceutical Sciences of CAAS (No. LIHPS-CAAS-2017-115) approve all experiments in this study. We took the blood samples of yaks in accordance with the guidelines for the care and use of laboratory animals.

### 2.2. Sample Collection, Sequencing and Sequence Alignment

The experimental subjects were 20 healthy, female, Tianzhu white yaks, including 10 long-haired yaks and 10 normal-haired yaks. All yaks were from the same breeding farm, and their feeding management and growth environment were consistent. Neck venous-blood samples were collected and stored in blood anticoagulant tubes at −20 °C for whole-genome resequencing.

Genomic DNA was extracted from the blood of Tianzhu white yaks using the blood Genome Extraction Kit (Tien Biotechnology, Beijing, China). The integrity and quality of the extracted DNA were examined using the Thermo Science NanoDrop 2000C (ThermoFisher Science Inc, Waltham, MA, USA) and 1.0% agarose-gel electrophoresis. The DNA library with 200–300 bp insertion fragments was constructed using the end-pairing method, and then the qualified DNA library was sequenced, based on the BGISEQ-500 platform. The clean reads after filtering out the low-quality reads were mapped to the Bos grunniens reference genome (LU_Bosgru_v3.0.105), using the BWA-MEM (0.7.10-r789) with default parameters.

### 2.3. Detection of Genome CNV Distribution

In this study, the Read Depth (RD) method was mainly used for genome-wide copy-number-variation detection, and the CNVnator (v0.3.2) software based on the RD method was used [24]. To decrease the rate of false-positive findings and obviate erroneous results, only CNVS larger than 0.5 KB were retained for further analysis [25,26].

### 2.4. Merging of CNV

The region of copy-number variation (CNVR) refers to the region formed by the combination of overlapping CNVS on the genome. By extending the boundary of overlapping CNVS, CNVs with 1 bp or more overlapped by different individuals were merged into CNVR [27]. CNVRs of 10 samples with the same phenotype were combined to obtain two VCF files containing the information for 10 individual CNVRs. In order to further decrease the false-positive detection rate, only CNVRs present in four or more samples were used for functional and comparative analysis, to minimize bias due to consistency in sequencing coverage [28,29].

### 2.5. Validation of CNVRs using Quantitative PCR (qPCR)

In order to test the accuracy of CNV allocation, quantitative PCR (qPCR) was used to verify the inferred CNVR. Six normal-haired and six long-haired individuals were randomly selected to verify 8 CNVRS. The qPCR primers were designed by the NCBI online design primer tool (https://www.ncbi.nlm.nih.gov/tools/primer-blast/, accessed on 29 August 2022). The bovine basic transcription factor 3 gene (BTF3) was selected as the internal reference gene. All primer sequences were synthesized by Tsingke Biotechnology (Beijing, China) and these primers are listed in Appendix A.

The real-time qPCR experiments were carried out in accordance with the manufacturer‘s instructions. The total reaction volume was 20 μL, containing 50 ng of template DNA, 10 mM primers, and reagents from SYBR Green Premix Pro Taq HS qPCR Kit (Accurate Biology, Hunan, China). All real-time reactions were performed using the LightCycler 96 Instrument (Roche, Basel, Switzerland), with 3 technical replicates per sample. Copy-number differences were determined using the standard ΔΔCT method.

### 2.6. Gene Annotation and Ontology

Based on the yak reference genome(Bos_grunniens.LU_Bosgru_v3.0.105, https://asia.ensembl.org/Bos_grunniens/Info/Index, accessed on 29 August 2022) and annotation files, a yak-genome annotation library was constructed using a local server, and then SnpEff (v4.5) was used to annotate the VCF files of the two analyzed populations. In accordance with the annotation information obtained, the intronic regions were removed to obtain meaningful variants and search for related genes. Then the functions of related genes were searched, through the gene database of the NCBI website(https://www.ncbi.nlm.nih.gov/, accessed on 29 August 2022), and the functional annotation of genes was carried out based on relevant literature reports. The gene-ontology (GO) enrichment analysis of the annotated genes was conducted using the online tool DAVID (https://david.ncifcrf.gov/, accessed on 29 August 2022). FDR was used to adjust the *p*-value, and the critical value of the adjusted *p*-value was set to 0.05.

## 3. Results

### 3.1. Resequencing Data Results

In this study, 20 individual samples of the white yak were resequenced, and the average sequencing depth was 7.48×. The statistical results showed that a total of 3,014,148,570 reads were obtained from 20 individuals, covering an average of 98.33 % of the reference genome (Table 1).

### 3.2. CNV Test Results

Genome-wide CNV detection of resequencing data using CNVnator software revealed a total of 110,268 CNVs in 20 samples, and these CNVs were composed of deletion and duplication mutation-events (Table 2, Figure 2). There were 53,099 CNVs in the long-haired samples, including 9634 duplicate events and 43465 deletion events, with an average of 5310 CNVs, 963 duplicate events, and 4347 deletion events, per individual. The size of CNVs ranged from 0.6 kb to 359.1 kb, with an average size of 13.9 kb and a median size of 9.9 kb. There were 57,169 CNVs in the normal-haired-type sample, including 8863 duplicate events and 48,306 deletion events, with an average of 5717 CNVs, 886 duplicate events, and 4831 deletion events, per individual. The size of CNVs ranged from 0.6 kb to 385.2 kb, with an average size of 13.2 kb and a median size of 9.3 kb. A total of 2006 CNVRs were obtained by combining CNVs from 10 long-haired individuals, and a total of 2699 CNVRs were obtained by combining CNVs from 10 normal-haired individuals. The accuracy of the identified CNVRS was confirmed using qPCR analysis. The results showed that 89% of the CNVS had accurate copy numbers (Appendix A).

### 3.3. Distribution of CNV on Chromosomes

Based on the CMplot package in the R language, we mapped the chromosomal distribution of CNV in the genomes of LTWY and NTWY (Figure 3). The distribution of CNV was found on each chromosome of LTWY and NTWY, but the distribution of CNV on each chromosome was not uniform, and the distribution density of CNV on each chromosome was not directly related to the length of chromosome.

### 3.4. Gene Annotation and Functional Analysis

SnpEff (V4.5) was used to annotate two VCF files containing CNV in the genome of LTWY and NTWY. A total of 103 genes were annotated in the long-haired-type file, 23 genes were annotated in the normal-haired-type file, and 80 genes were differentially annotated between the long-haired type and the normal-haired type (Appendix A). All genes annotated in the normal-haired-type population were included in the long-haired-type population. We carried out GO enrichment analysis of the differential genes, to understand their biological functions. The results showed that 80 differential genes were significantly enriched in 8 pathways (Table 3), (such as the glycerol-3-phosphate biosynthetic process, the regulation of the cholesterol metabolic process, the glycerol metabolic process, protein autophosphorylation, the DNA damage checkpoint, the triglyceride metabolic process, and the LINC complex). These pathways may affect the hair growth of the Tianzhu white yak. Moreover, it is worth noting that several differential genes (e.g., *ASTN2*, *ATM*, *COL22A1*, *GK5*, *SLIT3*, *PM20D1*, and *SGCZ*) have been identified as associated with hair growth or hair-follicle development in mammals.

## 4. Discussion

Hair length is a simple recessive-inheritance in animals. To reveal the potential contribution of CNV to hair growth in white yaks, we performed copy-number-variation analysis, based on whole-genome resequencing data from 10 long-haired and 10 normal-haired Tianzhu white yaks. We identified 2006 CNVRs and 130 genes in the long-haired type and 2699 CNVRs and 23 genes in the normal-haired type. The results showed that all the genes annotated from the normal-haired-type population were included in the long-haired-type population, indicating that the two populations were closely related. The Tianzhu white yak has been in a state of artificial breeding for a long time, and the long-haired type is a subgroup which emerged in a short time [3], and which is closely related to the normal-haired type. In addition, the results of the q-PCR showed that 89% of CNVs had an accurate copy-number. It is worth noting that not all CNVRs can be detected by qPCR, especially some low-copy repeats with low sequence-similarity [28,29].

In this work, a total of 80 differential genes were annotated, and they were the unique genes annotated in the long-haired type. These differential genes were significantly enriched in the GO terms of lipid synthesis and metabolism, and cell migration. Several studies have identified relationships between lipids and skin, hair follicles, and hair growth. Jiang et al. found that phospholipids and triglycerides affect the formation and function of the epidermal permeability barrier, and changes in lipid metabolism in the skin had a detrimental effect on the hair of mice [30]. Cholesterol has long been suspected of affecting hair biology, and Panicker et al. found that changes in cholesterol production within hair-follicle cells inhibited hair growth, eliciting immune responses and leading to hair loss [31]. Palmer et al. showed that dysregulation of cholesterol homeostasis is associated with several hair-growth and circulation disorders [32]. The LINC complex is a cell component enriched in our study. Studies have shown that the LINC complex is involved in the epidermal stem-cell differentiation, and affects the accessibility of epidermal differentiation genes [33]. In conclusion, these differential genes may affect hair growth, through lipid metabolism and other pathways.

In addition, we found that some of these differential genes have been identified as associated with hair growth in mammals. These genes, such as *GK5*, *ASTN2*, *ATM*, *COL22A1*, *SLIT3*, *PM20D1* and *SGCZ*, may be candidate genes affecting hair length of the Tianzhu white yak. The *GK5* gene is a skin-specific kinase primarily expressed in sebaceous glands, and *GK5* deficiency leads to elevated levels of lipids such as cholesterol, triglycerides, and ceramide in the skin [34]. Studies have shown that changes in lipid composition, including excessive production or accumulation of cholesterol precursors, can lead to changes in hair-follicle development [35]. In addition, the biosynthesis of lipids in the skin plays an important role in hair-follicle maintenance, hair-follicle morphogenesis, and the formation of the epidermal permeability barrier [36,37,38,39,40,41].

ATM is a serine/threonine kinase that not only participates as a major component in the DNA-damage-response (DDR) pathway, but also acts as an important sensor for reactive oxygen species (ROS) in cells, and is activated by oxidative stress [42,43,44]. Studies have shown that hair follicles have specific compartments of oxidative metabolism, and that the production of endogenous ROS is essential for many cell-signaling processes, including hair follicles, while oxidative stress induces hair-growth inhibition [45,46]. REDOX perception in a state of hair-follicle growth and differentiation with significant complexity and effect, and inhibition of *ATM* expression, promoted the oxidative-stress-induced loss of the activity of hair-follicle melanocytes [42], in which *ATM*, in protecting hair-follicle growth and differentiation from oxidative-stress damage can play a key role.

As a cell adhesion ligand for skin epithelial cells and fibroblasts, *COL22A1* shows a unique localization at the boundary between hair follicles and dermis in the skin during the growth phase, and is expressed around the lower third of hair follicles in the skin, during the growth phase [47]. Several studies have shown that the synergistic effect of TGF-β and Wnt signaling plays a crucial part in controlling some developmental events, especially hair-follicle formation [48,49,50,51]. Studies have found that *COL22A1* is a top regulatory gene regulating the TGF-β pathway [52]. Therefore, *COL22A1* may affect hair-follicle development during the growth phase by affecting the the synergistic effect of the TGF-β and Wnt signaling-pathways, leading to inconsistent hair growth.

The *ASTN2* gene is an integrated membrane glycoprotein unique to vertebrates [53]. Planar-cell-polarity signals (PCP signals) not only have a major impact on many developmental processes, but also control the orientation of mammalian hair follicles [54]. It has been reported that *ASTN2* is expressed in various tissues during development, and begins to be expressed at the earliest stage of hair-follicle development; studies on its endosomal localization have found that it may play a role in the recycling of plasma membrane proteins [55,56]. Researchers have found that the PCP-protein complexes in the developing epidermis are recycled, assembled and utilized at the plasma membrane, as cells divide [57]. These results suggest that alterations in *ASTN2* may affect hair-follicle development by affecting PCP-protein transport. Therefore, we hypothesized that the CNV of the *ASTN2* gene may affect PCP-protein transport and PCP signaling, thus affecting the hair follicles of the Tianzhu white yak. *SLIT3*, a collagen regulator secreted by fibroblasts, mainly exists in fibrous collagen-producing cells, and negatively regulates cell growth [58,59]. Studies have reported that, compared with normal mice, *SLIT3*-knockout mice have significantly lower hair-follicle density [58], and some researchers have found that *SLIT3* is associated with wool traits of fine wool sheep [60]. *PM20D1* encodes a protein of unknown function, but studies have shown that its expression affects the growth of human sebaceous glands and mouse hair follicles [61,62]. *SGCZ* has been detected to be involved in fat deposition and hair growth in sheep [63,64]. All these genes are related to hair growth or hair-follicle development, and may be candidate genes affecting the hair length of the Tianzhu white yak.

## 5. Conclusions

This is the first analysis of CNV in the whole genome of the Tianzhu white yak. By comparing the CNV of long-haired with that of normal-haired Tianzhu white yaks, we found that some genes which overlapped with CNVR may be candidate genes for affecting hair growth in Tianzhu white yaks. This study provides valuable insights for phenotypic variation and breeding of the Tianzhu white yak.

## Figures and Tables

**Figure 1 genes-13-02405-f001:**
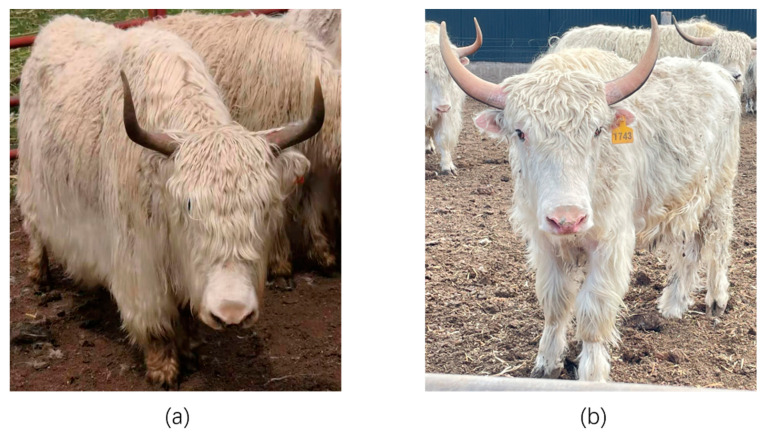
(**a**) Long-haired Tianzhu white yak; (**b**) Normal-haired Tianzhu white yak.

**Figure 2 genes-13-02405-f002:**
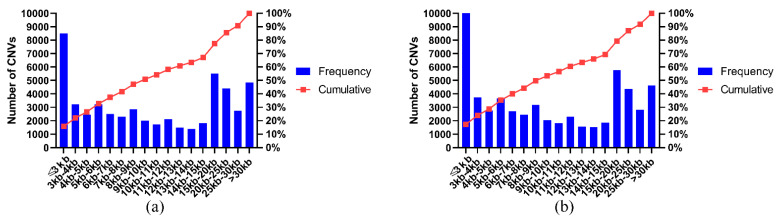
(**a**) CNV-size interval distribution of LTWY. Average CNV size is 13.9 kb and median size is 9.9 kb; (**b**) CNV-size interval distribution of NTWY, Average CNV size is 13.2 kb and median size is 9.3 kb.

**Figure 3 genes-13-02405-f003:**
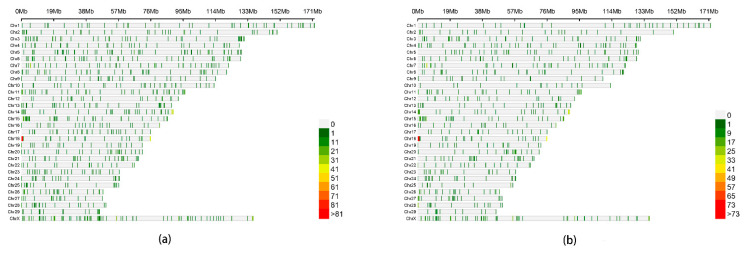
(**a**) Density of CNVRs per chromosome in LTWY; (**b**) density of CNVRs per chromosome in NTWY.

**Table 1 genes-13-02405-t001:** Summary statistics of resequencing reads.

Sample Name	Number	Raw Reads	Mapped Reads	Properly Paired Reads	Average Coverage	Average Fold
Long-haired	10	1,520,934,131	1,495,623,348	1,409,965,496	98.33%	7.38×
Normal-haired	10	1,544,406,470	1,518,525,222	1,428,802,730	98.33%	7.58×
Total	20	3,065,340,601	3,014,148,570	2,838,768,226	98.33%	7.48×

**Table 2 genes-13-02405-t002:** Summary of CNV testing.

Sample Name	Number	CNVs	Duplication	Deletion	Individual Average CNVs
Long-haired	10	53,099	9634	43,465	5310
Normal-haired	10	57,169	8863	48,306	5717
Total	20	110,268	18,497	91,771	5513

**Table 3 genes-13-02405-t003:** The significant GO categories of CNVR-harbored genes.

GO ID	Function	GO Type	Adjusted *p*-Value	Name of CNV Harbored Genes
GO:0021817	nucleokinesis involved in cell motility in cerebral-cortex radial glia-guided migration	biological process	0.010975072	*SUN1*, *SYNE2*
GO:0046167	glycerol-3-phosphate biosynthetic process	biological process	0.018225877	*GK5*, *FGGY*
GO:0090181	regulation of cholesterol metabolic process	biological process	0.036124919	*EPHX2*, *TTC39B*
GO:0006071	glycerol metabolic process	biological process	0.039666016	*GK5*, *FGGY*
GO:0034993	LINC complex	cellular component	0.032183984	*SUN1*, *SYNE2*
GO:0005887	integral component of plasma membrane	cellular component	0.063910999	*TSPAN15*, *INSR, TSPAN18*, *PCDH15, TAS1R2*, *HTR3B, ASIC2*
GO:0004370	glycerol kinase activity	molecular function	0.015268554	*GK5*, *FGGY*
GO:0016773	phosphotransferase activity, alcohol group as acceptor	molecular function	0.033288799	*GK5*, *FGGY*

## Data Availability

The bioproject number of the sequencing data information about long-haired Tianzhu white yak and normal-haired Tianzhu white yak is PRJNA766811 in the NCBI Sequence Read Archive.

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
