# Peer review of "Analysis of Copy Number Variation in the Whole Genome of Normal-Haired and Long-Haired Tianzhu White Yaks"

_genes, 2022, doi:10.3390/genes13122405_

Round 1

Reviewer 1 Report

Yak hair is one of its most valuable resource and longer hairs are of greater economic utility for the farmers. The study has been well conceived with a sound methodology and presents some interesting insights on hair growth pattern in two sub-populations of Tianzhu white yaks. However, some points need to be clarified which are as follows:

1.      Are the two sub-groups subjected to shearing practices?

2.      Can you rule out the influence of sex and season in trait selection? Please provide some clarity on this aspect in the methodology

3.      Is the habitat, production systems and distribution of the two sub-groups same? Provide the details in introduction or methodology part accordingly

4.      Is Lu_bosgru_V3.0 reference genome available online? If so, Please provide reference

5.      Provide qPCR results for clarity on the validation part. Some pictures for differential expression results would be highly desirable

6.      Insert pictures of the long-haired and normal-haired Tianzhu white yaks for better visual appraisal by the readers

7.      What was the threshold FDR or p-value considered for enrichment analysis or significance? Please mention

8.      It is not desirable to put references in the results section. Please correct accordingly

Reviewer 2 Report

The article sent for reviewing has analysed the CNV differences in 20 whole genome sequences among 10 normal and 10 long hair Tianzhu White Yaks samples.

From a general point of view the article could be of interest for GENES readers but in my opinion it is neccesary a major revision.

1.- INTRODUCTION

A better description of demographic data should be included in the introduction. Authors comment that the census of short-haired White yaks is large but "some" long-haired inidividuals have been recently found. "SOME" it is a very inespecific parameter, a better demographic description should be included in the article, how much? is it a specific event in few herds? farmers select against or favouring long haired phenotypes?. Long-haired phenotype it is a desirable trait for farmers and will replace short-haired phenotypes?

2.- MATERIAL AND METHODS

Line 75 How was the sampling procedure? the sample was randomly selected? adn the remainig mangament conditions tha feed were different?

There are some methods widely described while others not. In my opinion The genomic extraction and sequencing procedure could be summarize (line 78 to 94), but it is not commnet anything about the sequence alignment methodology. Furthermore a little description has been included in the results section (Li37-139) thta should be included in material and methods section.

Line 102 What is the meaning of "an arbitrary amonut of overlap"? One base, more than 100 basess,...?

Line 106 Why the authors have choosen 4 or more samples as the limit to Include/exclude CNVR? Is it an arbitrary criteria?

Line 115 I did not find Table S1 among the documents uploaded.

Line 111 Why the CNVR have been verified in 12 samples and not in all the samples? Which 12 samples were analysed? Please explain the total of long and short-haired samples are included among the 12 samples.

Also, the results section described that a ttoal of 2699 CNVR were identified, but 8 were verified (line 111). Please explain better this methodology.

3.- RESULTS

Lin 155 What is the meaning of "trusted CNVR"?

Line 178-180 A supllemtary table with the 103 geens description should be inlcuded as suplementary material.

Line 184 What is it the meaning of "80 total concentration"?

4.- DISCUSSION

Line 220 The list of genes described are not mentioned in the "name of CNV harbored genes" included in table 3, except for GK5. So, were not selected from the results of the analysis? were selected from the bibliography? Please clarify this issue.

Do the auhors think that the number of sample analysed are enough to this kind of analysis? Do the results could be influenced by other characteristics than Long/Short hair?

Best regards.

Author Response

请参阅附件。
